# Development and validation of multivariable machine learning algorithms to predict risk of cancer in symptomatic patients referred urgently from primary care: a diagnostic accuracy study

Richard Savage [iD],[1] Mike Messenger,[2,3] Richard D Neal [iD],[2,3,4] Rosie Ferguson,[1] Colin Johnston,[5] Katherine L Lloyd,[1] Matthew D Neal,[1] Nigel Sansom,[1] Peter Selby,[2,3,6] Nisha Sharma,[5] Bethany Shinkins,[2] Jim R Skinner,[1] Giles Tully,[1] Sean Duffy,[5] Geoff Hall[2,3,5]

RS, MM and RDN are joint first authors.
SD and GH are joint last authors.

For numbered affiliations see end of article.

**Correspondence to**
Dr Richard Savage;
rich.savage@
pinpointdatascience.com

## ABSTRACT

**Objectives** To develop and validate tests to assess the risk of any cancer for patients referred to the NHS Urgent Suspected Cancer (2-week wait, 2WW) clinical pathways.
**Setting** Primary and secondary care, one participating regional centre.
**Participants** Retrospective analysis of data from 371 799 consecutive 2WW referrals in the Leeds region from 2011 to 2019. The development cohort was composed of 224 669 consecutive patients with an urgent suspected cancer referral in Leeds between January 2011 and December 2016. The diagnostic algorithms developed were then externally validated on a similar consecutive sample of 147 130 patients (between January 2017 and December 2019). All such patients over the age of 18 with a minimum set of blood counts and biochemistry measurements available were included in the cohort.
**Primary and secondary outcome measures** sensitivity, specificity, negative predictive value, positive predictive value, Receiver Operating Characteristic (ROC) curve Area Under Curve (AUC), calibration curves
**Results** We present results for two clinical use-cases. In use-case 1, the algorithms identify 20% of patients who do not have cancer and may not need an urgent 2WW referral. In use-case 2, they identify 90% of cancer cases with a high probability of cancer that could be prioritised for review.
**Conclusions** Combining a panel of widely available blood markers produces effective blood tests for cancer for NHS 2WW patients. The tests are affordable, and can be deployed rapidly to any NHS pathology laboratory with no additional hardware requirements.

## BACKGROUND

A major National Health Service (NHS) cancer policy to diagnose cancer earlier led to the introduction of Urgent Suspected Cancer referrals. These referrals are predicated

### Strengths and limitations of this study

► It is based on well-validated, low-cost clinical assays already available at scale in NHS pathology laboratories; the tests could therefore be deployed across the UK very rapidly, with no additional hardware requirements.
► The large numbers of cases reported, and that the performance estimates are conservative due to missing data and the historical nature of the blood measurements; prospective evaluation will not suffer from these drawbacks. The principal limitations of this work are:
► That the development and validation was done only in one centre.
► There is a possible source of bias, in that the subset of patients who had retrospective blood data may not be representative of the overall 2-week wait cohort.
► We have only reported the validation on a retrospective sample; a prospective evaluation is needed.

on the risk of symptomatic patients having cancer.[1] Trusts assess patients within 2 weeks ('2-week wait' (2WW) referral). The 2WW pathways have contributed to improving outcomes; higher general practice use of referrals for suspected cancer is associated with lower mortality for the four most common types of cancer (prostate, breast, lung and colorectal).[2]

This approach places a major strain on diagnostic services on NHS England, with over 2 million 2WW referrals annually, and a 10% year-on-year increase in referrals over the past decade.[3] This highlights an unsustainable

burden on existing services, workforce and financial resources. While there is variation between cancer pathways, only 7% overall of 2WW referral patients are diagnosed with cancer.[3] Many patients are therefore subject to unnecessary psychological distress, as well as being exposed to diagnostic tests which may inadvertently cause harm. Clearly there is a need to improve the efficiency of these pathways.

These challenges are exacerbated by the current COVID-19 crisis. The NHS capacity to assess 2WW referrals is reduced, and a backlog of referrals continues to build.[3 4] These unprecedented challenges urgently require new solutions. COVID-19 has presented an opportunity for general practitioners (GPs) to permanently change how they use emerging technologies.[5]

Many biomarkers have been evaluated for their use in cancer diagnosis; however, only a few are currently used in either primary or secondary care settings. A systematic mapping review identified 94 ctDNA studies alone, highlighting how much more work is required prior to clinical use.[6] Companies like GRAIL and Freenome are pursuing this, with clinical trials ongoing.[7 8] There is also evidence that signals from a range of different analytes can be usefully combined via machine learning.[9]

Using such approaches to triage cancer referrals should bring benefits to patients, health systems and the economy. For example, a rule-out test for symptomatic patients, like those referred to the NHS 2WW, could identify those with very low cancer risk, allowing many patients without cancer to avoid unnecessary procedures and freeing up diagnostic capacity for those at greater risk.

The work presented in this paper addresses the top three priority areas identified by Badrick *et al*, including: a simple, non-invasive, painless and convenient test to detect cancer early; a blood test to detect some or all cancers early that can be included into routine care; and a test that is easily accessible to general practice.[10]

We report the development and validation of a set of machine learning algorithms to provide a calibrated risk probability of cancer (a score between 0 and 1, higher values indicating greater risk of cancer) for triaging symptomatic patients. A calibrated risk probability has a variety of clinical uses. This paper focuses on the two use-cases for the NHS 2WW:

Use-case 1—a rule-out test when patient has a very low risk of cancer, allowing initial management in primary care.

Use-case 2—a way of identifying patients at high risk of having cancer to fast-track them for further tests.

## METHODS
### Methodological design and source of data
This work is a single centre, retrospective diagnostic prediction study (classified as a type 2b study by the transparent reporting of a multivariable prediction model for individual prognosis or diagnosis (TRIPOD) statement.[11] The prediction algorithms were developed and validated on a large data set from a single geographic area, split chronologically into two independent cohorts.

The data set contained 371 799 consecutive 2WW referrals in the Leeds region from 2011 to 2019. The development cohort was composed of 224 669 consecutive patients with an urgent suspected cancer referral in Leeds between January 2011 and December 2016. The diagnostic algorithms developed were then externally validated on a similar consecutive sample of 147 130 patients (between January 2017 and December 2019). Both development and validation sets were selected using the same inclusion and exclusion criteria and both received the same preprocessing, consisting of removing greater than ('>') symbols from blood analyte values in the data, and setting data values with less than ('<') values to zero. This is a simple imputation for the case where a pathology laboratory returns a result outside the reportable range. Because the chosen machine learning algorithms are not sensitive to scaling of individual variables, it was not necessary to normalise the inputs.

### Participants
Patients were selected because they received a 2WW referral to Leeds Teaching Hospitals NHS Trust (LTHT) during the above time frame. Referrals were included for all 2WW pathways, and all patients over the age of 18 with a minimum set of blood counts and biochemistry measurements available were included in the cohort. Occasional multiple referrals of the same patient (for example to different 2WW pathways) is expected in this data set— such instances are infrequent, and are not modelled any differently from other referrals. While information about repeated referral could, in principle, aid the algorithm, this would make the algorithm much harder to deploy in practice as it would need reliable access to an electronic healthcare record, rather than just being linked directly to the laboratory information management system, which handles the pathology lab data flows. We have, therefore, avoided this on practical grounds, for the time being.

Patients from all 2WW pathways were included in the development set; patients from the nine 2WW pathways at LTHT considered in this paper were included in the validation set. The reason for including all cases in the development set is that our goal was to train algorithms that could assist with pan-cancer diagnosis, including cancer cases which have not been referred down the correct pathway. Validation was restricted to these nine 2WW pathways (which account for ~98% of all 2WW referrals in England) because the remaining pathways, being much smaller did not have sufficient validation data to provide useful validation. Patients not fulfilling these criteria were excluded from the analysis. All patients were followed up to 12 months after the conclusion of their referral, or until February 2020. Patients in the validation set (ie, referred from January 2017 onwards) only required the outcome of the 2WW referral and therefore the possibility of censoring of outcomes up to 12 months did not affect the validation results.

**Table 1** Total number of cases per pathway (2011–2019)

| Pathway | 2011–2016 | 2017–2019 | Total |
|---|---|---|---|
| Breast | 60 673 | 36 561 | 97 234 |
| Lower Gastrointestinal | 31 966 | 22 331 | 54 297 |
| Upper Gastrointestinal | 18 986 | 11 938 | 30 924 |
| Gynaecological | 16 533 | 11 599 | 28 132 |
| Urological | 20 209 | 13 326 | 33 535 |
| Lung | 7607 | 3237 | 10 844 |
| Haematological | 2273 | 1323 | 3596 |
| Head and neck | 22 594 | 14 558 | 37 152 |
| Skin | 38 605 | 29 239 | 67 844 |
| Key pathways total | 219 446 | 144 112 | 363 558 |
| All pathways total | 224 669 | 147 130 | 371 799 |

## Outcome

The algorithms were trained to predict whether or not a patient would receive a cancer diagnosis. Outcome labels were derived from International Classification of Disease (ICD-10) diagnostic codes from the Leeds secondary care cancer clinical database. 'Cancer' was defined as any patient diagnosed with a malignant (ICD-10 'C' codes) or in situ (appropriate subset of ICD-10 'D' codes) neoplasm as the result of their referral or within the subsequent 12-month period for the purposes of model development. Diagnoses as the result of an urgent referral were used as outcomes in the validation analyses, to match the intended clinical setting. Benign neoplasms were defined as 'Not Cancer'. The full list of ICD-10 codes designated as 'cancer' are in online supplemental materials.

## Predictors

The variables for each patient include a full blood count, a range of biochemistry measurements, a panel of standard tumour markers, plus age and sex. All predictors were included on their natural scale (ie, they were not normalised or dichotomised).

As a retrospective cohort, blood measurements were used where they were available in the database up to 90 days prior to referral or up to 14 days post referral. This was done to seek a reasonable balance between missing data and possible bias (eg, if blood measurements were made after a diagnosis had been established). For example, it is risky to use blood measurements taken more than 14 days post-referral as there is an increasing chance that those bloods could have been ordered by a clinician in response to a confirmed diagnosis of cancer. In routine clinical use, all model predictors would be available at the time.

## Sample size

The protocol for this work stated a goal of achieving a negative predictive value (NPV) of 0.99 or greater for the rule-out use-case. Because NPVs below 0.99 are undesirable, we consider sample sizes as they impact the lower half of the 95% CI for NPV. For a 0.05 lower CI size, we require 100 total patients being ruled out; for a 0.02 lower CI size we require 300 patients. With a design goal of achieving a 20% rule-out rate, this would therefore require approximately (100)/(0.2)=500 total cases per pathway for a 0.05 lower CI size, or (300)/(0.2)=1500 total cases per pathway for a 0.02 lower CI size.

The validation set meets the above sample size criteria for 7 of the 9 2WW pathways for which results are presented. The other two pathways (lung and haematological) are high prevalence pathways (see tables 1 and 2), and so it was decided to also include results for these two pathways as the 95% CI are provided for all results to make clear the level of uncertainty present due to sample sizes. The remaining (smaller) 2WW pathways as recorded in the clinical data were also considered (testicular, brain/Central Nervous System (CNS), sarcomas,

**Table 2** Number of cases meeting bloods criteria

| Pathway | Development set | | | Validation set | | |
|---|---|---|---|---|---|---|
| | # Cancer | # Non-cancer | Prevalence | # Cancer | # Non-cancer | Prevalence |
| Breast | 807 | 7571 | 9.6 | 424 | 5219 | 7.5 |
| Lower Gastrointestinal | 1257 | 11 401 | 9.9 | 856 | 9361 | 8.4 |
| Upper Gastrointestinal | 662 | 5317 | 11.1 | 428 | 4337 | 9.0 |
| Gynaecological | 407 | 3098 | 11.6 | 218 | 2278 | 8.7 |
| Urological | 1836 | 4677 | 28.2 | 1143 | 3063 | 27.2 |
| Lung | 687 | 1380 | 33.2 | 177 | 616 | 22.3 |
| Haematological | 403 | 654 | 38.1 | 180 | 343 | 34.4 |
| Head and neck | 546 | 4293 | 11.3 | 346 | 3177 | 9.8 |
| Skin | 1468 | 3910 | 27.3 | 1287 | 3427 | 27.3 |

Details of the cases which meet the acceptance criteria for the analyses presented in this paper. Prevalence is calculated only for those cases meeting the criteria, and not for all patients entering a given pathway.

children's cancer, acute leukaemia, other cancer), but we did not develop algorithms for these as the available sample sizes were judged too small to train and validate effective models.

## Management of missing data

Missing data is a key issue for this cohort as many patients did not have bloods in this time frame (see tables 1 and 2). Patients were identified who had full blood counts and a minimum subset of biochemistry data, and this subset was used to train the algorithms. The core algorithms use a gradient boosting model including an inbuilt method for imputing missing data which infers from the data how to handle missing data values, by learning at each decision tree node in the ensemble which branch a missing value should be assigned to. Early work during model development showed that this inbuilt method modestly outperformed (in a statistical sense) simple imputation methods, and has the advantage of simplifying the model development somewhat.

## Patient and public involvement

Multiple public and patient consultations have been undertaken in relation to this work, initially via the NIHR-Leeds In Vitro Diagnostics Co-Operative (Leeds MIC) Public and Patient Interaction/Engagement group, expanding to Healthwatch Leeds and Healthwatch Kirklees as well as the West Yorkshire and Harrogate Cancer Alliance and CANTEST programme patient panels. Several sessions have been held and feedback gained on the clinical use of the tests presented in this work.

## Statistical analysis methods

The goal of the algorithms is to produce a well-calibrated prediction of the probability that a patient has cancer. The type of model required is a probabilistic classifier—a model that predicts the probabilities of a given patient belonging to one of several distinct classes.

The development set was used to identify appropriate models and calibration methods and to tune the hyperparameters for those models. Methods and hyperparameters were compared and tuned using fivefold cross-validation. This was concluded and results locked down before validation.

The model structure selected using the development set is a combination of a core machine learning algorithm with good predictive performance (gradient boosting), plus a calibration step (polynomial logistic regression, a modified version of Platt Scaling.[12] Gradient boosting was chosen for a number of pragmatic and statistical performance reasons. It is generally seen to perform very well in comparison to other methods on structured data sets such as are used in this paper and we observed the same thing during early development work. Gradient Boosting using decision trees is also able to straightforwardly handle input variables with wildly different distributions (eg, tumour markers vs blood counts). There are several very good Python packages available that implement gradient

boosting (we use XGBoost[13] and LightGBM,[14] and these packages have built-in methods for handling missing data. Gradient boosting also has a modest computational load for both training and prediction. Platt Scaling is a standard calibration method which uses logistic regression. We have modified this to use polynomial logistic regression because we found this gave better calibration performance with the outputs of our gradient boosting algorithms.

The outcome classes for this work are significantly imbalanced, with substantially fewer cancers than non-cancers (see prevalences in table 2). The imbalanced classes are accounted for via upweighting the importance of the cancer patients in the gradient boosting algorithms. The same weight is applied to all cancer patients, and this is tuned as a hyperparameter during the development work (ie, using cross-validation on the development set).

Prior to any analysis variables were selected based on: cost and relevance, availability in NHS pathology labs and prior knowledge from medical literature that they might reasonably be expected to contain some cancer-relevant information. Variable selection in the statistical sense (ie, using the development data set) was not carried out and the gradient boosting algorithm used in this work is able to down-weight any input variables which are of lesser statistical importance (in terms of contribution to making good predictions).

The validation set was used to validate the locked-down algorithms. After this no changes were made to the algorithms, results are presented below.

## RESULTS

Figure 1 shows a Consolidated Standards of Reporting Trials (CONSORT) flow diagram for this work.

Tables 1 and 2 show the total number of cases per pathway, and the number of those cases meeting the inclusion criteria. Tables 3 and 4 show the age and sex demographics of the included patients, by pathway and by development/validation set.

Table 5 shows test performance characteristics for nine urgent referral pathways for use-case 1 (rule-out). The goal here is to successfully identify 20% of non-cancer patients (a specificity of 0.2) who are at very low risk of cancer, so that other possible causes of their symptoms can be considered rather than continuing with a 2WW referral.

Table 6 shows test performance characteristics for use-case 2 (triage), to identify patients at higher risk of cancer who would be considered for priority through the urgent referral pathway. The goal here is to successfully red-flag 90% of cancer cases (a sensitivity of 0.9) for priority investigation.

Figure 2 shows an example of stratification via a test, compared with the existing standard care pathway. In this example, 500 patients present to the breast pathway, which is overloaded and only able to see 400 of these patients within 2 weeks of their referral. The standard care pathway is modelled as first-come first-served, and so the proportion of patients with cancer is the same in

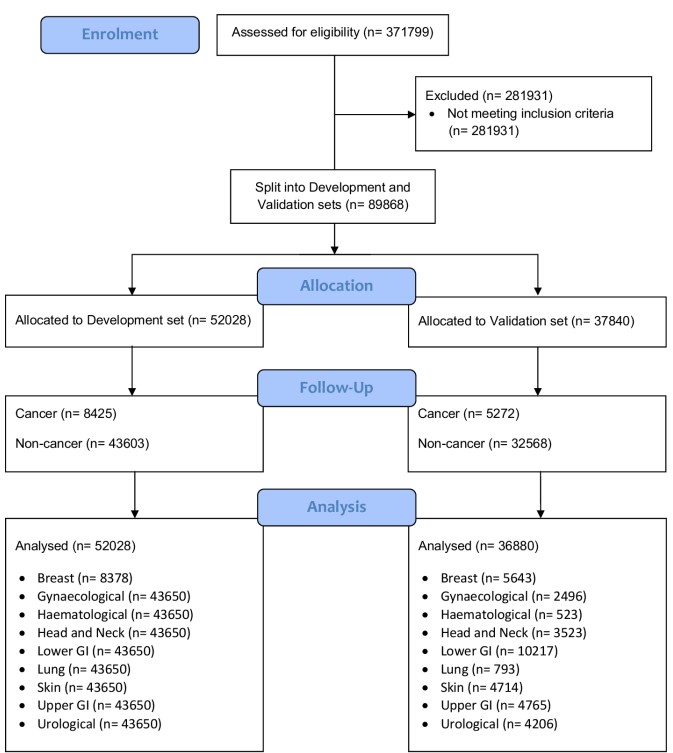

Figure 1 showing CONSORT flow diagram:

- Enrolment
  - Assessed for eligibility (n= 371799)
  - Excluded (n= 281931)
    - Not meeting inclusion criteria (n= 281931)
  - Split into Development and Validation sets (n= 89868)
- Allocation
  - Allocated to Development set (n= 52028)
  - Allocated to Validation set (n= 37840)
- Follow-Up
  - Cancer (n= 8425) / Non-cancer (n= 43603)
  - Cancer (n= 5272) / Non-cancer (n= 32568)
- Analysis
  - Analysed (n= 52028)
    - Breast (n= 8378)
    - Gynaecological (n= 43650)
    - Haematological (n= 43650)
    - Head and Neck (n= 43650)
    - Lower GI (n= 43650)
    - Lung (n= 43650)
    - Skin (n= 43650)
    - Upper GI (n= 43650)
    - Urological (n= 43650)
  - Analysed (n= 36880)
    - Breast (n= 5643)
    - Gynaecological (n= 2496)
    - Haematological (n= 523)
    - Head and Neck (n= 3523)
    - Lower GI (n= 10217)
    - Lung (n= 793)
    - Skin (n= 4714)
    - Upper GI (n= 4765)
    - Urological (n= 4206)

(Diagram adapted from CONSORT 2010 flow diagram, http://www.consort-statement.org/consort-statement/flow-diagram)

**Figure 1** We note that the development set analysed numbers of data points (bottom left) are the same for all pathways with the exception of breast. we discovered during development that modest performance gains could be achieved by using just the 2WW breast pathway data for the breast algorithm, and using the data for all other pathways for each of the other eight algorithms (hence the same training data were used for all pathways except breast). 2WW, 2-week wait. GI, Gastrointestinal.

the patients seen and the patients not seen. Using the test for stratification, the patients are stratified into high-risk, medium-risk and low-risk groups. Patients are then seen in risk order—in this example, all of the high-risk patients are seen, and some of the medium-risk patients are seen. Under stratification, far more of the patients with cancer are seen, and of the patients not seen, a far smaller proportion have cancer. An interactive version of this is available at https://www.pinpointdatascience.com/patient-test-stratification.

## DISCUSSION
### Summary of main findings
The NHS 2WW pathways are a major route through which symptomatic patients in the UK are assessed for possible cancer diagnoses. These pathways have been very successful in helping contribute to earlier cancer detection, but the number of 2WW referrals has doubled over the last decade and this has placed a major strain on diagnostic services. These challenges have been exacerbated by the current COVID-19 crisis, with the NHS capacity to assess 2WW referrals reduced, and a backlog of referrals continuing to build.

New diagnostic technologies have the potential to play a role in solving this challenge. This paper reports the development and validation of a set of statistical machine learning algorithms based on routine laboratory blood measurements that can predict cancer outcomes for symptomatic patients referred urgently from primary care for possible cancer diagnosis.

Each algorithm is trained and validated as a test to provide decision support for one of the nine NHS 2WW pathways. Each test produces a calibrated probability that the patient on that 2WW pathway has any type of cancer. These calibrated probabilities can be used in a range of clinical contexts; in this paper we consider two principal use-cases. In use-case 1, the tests are used to rule-out patients whose risk of cancer is very low, allowing clinicians to identify patients for whom investigations of possible non-cancer causes of their symptoms might be more appropriate. In use-case 2, higher-risk patients are red-flagged so that their onwards journey through the 2WW pathway can be expedited.

The main findings of this work are that it is possible to combine a panel of widely available blood markers to produce effective blood tests for cancer for NHS 2WW patients. Such tests are affordable, and can be deployed rapidly to any NHS pathology laboratory with no additional hardware requirements.

### Discussion of main findings within the context of the literature
This work is novel, innovative, and potentially of huge importance for the management of patients referred urgently for suspected cancer. The tests are based on a panel of routine blood measurements that: are already in common usage in NHS laboratories; work across a range of cancers; can easily be integrated with existing NHS systems. The tests have already been integrated with Mid-Yorkshire Hospitals NHS Trust Laboratory systems.

The tests can both identify patients at higher risk of cancer, such that they can be prioritised for assessment and diagnostic investigations, while also identifying a significant proportion of patients at very low risk who may not need further investigation for suspected cancer. Patients in both groups stand to benefit, either from expedited testing, or from not being exposed to iatrogenic harm and unnecessary cancer worries. The tests can be set at different thresholds in different cancers and within different health settings, making them responsive to local needs, capacity and priorities. COVID-19 has reduced diagnostic capacity and efficiency, this test could be an effective and rapid solution at this time of crisis.

An important practical note is that the criteria for 2WW changed in 2015, reducing the risk threshold warranting an urgent referral from 5% positive predictive value (PPV) to 3% PPV (ie, towards the end of the development cohort time frame). The validation results therefore

**Table 3** Age demographics

| Pathway | Development set | | | Validation set | | |
| --- | --- | --- | --- | --- | --- | --- |
| | Age 25th percentile | Age median | Age 75th percentile | Age 25th percentile | Age median | Age 75th percentile |
| Breast | 36 | 48 | 64 | 35 | 48 | 62 |
| Lower Gastrointestinal | 59 | 69 | 78 | 59 | 69 | 78 |
| Upper Gastrointestinal | 57 | 68 | 77 | 55 | 67 | 76 |
| Gynaecological | 49 | 57 | 69 | 46 | 54 | 66 |
| Urological | 58 | 68 | 77 | 59 | 69 | 78 |
| Lung | 58 | 69 | 78 | 57 | 67 | 76 |
| Haematological | 43 | 63 | 76 | 43 | 62 | 75.5 |
| Head and neck | 47 | 60 | 72 | 47 | 59 | 72 |
| Skin | 52 | 69 | 80 | 52 | 69 | 80 |

encompass this change in clinical practice, suggesting a certain robustness to those results.

### Strengths

This work is based on well-validated, low-cost clinical assays (see online supplemental table S5) already available at scale in NHS pathology laboratories. The tests could, therefore, be deployed across the UK very rapidly, with no additional hardware requirements. These tests are CE marked and are currently undergoing service evaluation in the West Yorkshire and Harrogate Cancer Alliance. The use of low-cost assays means that these tests are very affordable in comparison to typical per-patient 2WW referral costs.[15]

The performance estimates are conservative due to missing data and the historical nature of the blood measurements; prospective evaluation will not suffer from these drawbacks. Even biomarkers with limited individual performance are of value in this approach if they contribute complementary information. The algorithms are designed to be flexible, allowing thresholds to be changed according to clinical need, for example, use-case 2 during the COVID-19 pandemic. The large numbers reported, the robust analysis and reporting in line with TRIPOD and PROBAST.[11 16] There is the potential to improve performance using the pipeline of new biomarkers being developed for diagnostic, predictive or prognostic purposes.

### Limitations

The development and validation was done only in one centre, although a large regional cancer centre. We have also only reported the validation on a retrospective sample—a prospective multicentre evaluation is needed to provide confidence in the generalisability of the model.

We note that the validation set meets the defined sample size criteria (1500 total cases) for 7 of the 9 2WW. 95% CI are provided for all results to make clear the level of uncertainty present due to sample sizes. The remaining (smaller) 2WW pathways as recorded in the clinical data were also considered (testicular, brain/CNS, sarcomas, children's cancer, acute leukaemia, other cancer), but we did not develop algorithms for these as the available sample sizes were judged too small to train and validate effective models.

There is a possible source of bias, in that the subset of patients who had retrospective blood data may not

**Table 4** Sex demographics

| Pathway | Development set | | Validation set | |
| --- | --- | --- | --- | --- |
| | # Female (%) | # Male (%) | # Female (%) | # Male (%) |
| Breast | 7345 (87.67) | 1033 (12.33) | 5146 (91.19) | 497 (8.82) |
| Lower Gastrointestinal | 6889 (54.42) | 5769 (45.58) | 5529 (54.12) | 4688 (45.88) |
| Upper Gastrointestinal | 3346 (55.96) | 2633 (44.04) | 2746 (57.63) | 2019 (42.37) |
| Gynaecological | 3505 (100.00) | 0 (0.00) | 2495 (99.96) | 1 (0.04) |
| Urological | 1700 (26.10) | 4813 (73.90) | 904 (21.49) | 3302 (78.51) |
| Lung | 947 (45.82) | 1120 (54.19) | 363 (45.78) | 430 (54.22) |
| Haematological | 506 (47.87) | 551 (52.13) | 227 (43.40) | 296 (56.60) |
| Head and neck | 2755 (56.93) | 2084 (43.07) | 2080 (59.04) | 1443 (40.96) |
| Skin | 2924 (54.37) | 2454 (45.63) | 2614 (55.45) | 2100 (44.55) |

**Table 5** Twenty per cent rule-out

| Pathway | Proportion of non-cancers ruled-out (specificity) (95% CI) | Negative predictive value (95% CI) | Sensitivity (95% CI) |
|---|---|---|---|
| Breast | 0.2036 (0.1926 to 0.2143) | 0.9936 (0.9883 to 0.9981) | 0.9776 (0.9596 to 0.9933) |
| Lower Gastrointestinal | 0.2002 (0.1921 to 0.2081) | 0.9823 (0.9762 to 0.9877) | 0.9348 (0.9135 to 0.9543) |
| Upper Gastrointestinal | 0.2017 (0.1901 to 0.2137) | 0.9880 (0.9806 to 0.9946) | 0.9580 (0.9323 to 0.9804) |
| Gynaecological | 0.2040 (0.1871 to 0.2209) | 0.9895 (0.9799 to 0.9979) | 0.9718 (0.9462 to 0.9942) |
| Urological | 0.2002 (0.1864 to 0.2141) | 0.9525 (0.9358 to 0.9680) | 0.9681 (0.9568 to 0.9785) |
| Lung | 0.2031 (0.1704 to 0.2331) | 0.9630 (0.9281 to 0.9924) | 0.9673 (0.9364 to 0.9933) |
| Haematological | 0.2095 (0.1694 to 0.2542) | 0.9375 (0.8795 to 0.9868) | 0.9697 (0.9408 to 0.9938) |
| Head and neck | 0.2001 (0.1862 to 0.2139) | 0.9748 (0.9623 to 0.9858) | 0.9267 (0.8917 to 0.9580) |
| Skin | 0.2002 (0.1868 to 0.2130) | 0.9406 (0.9232 to 0.9570) | 0.9609 (0.9493 to 0.9717) |

be representative of the overall 2WW cohort. Different pathways have different conventions as to what blood tests are performed as part of a 2WW referral. For example, we note that the proportion of men with a breast 2WW referral meeting the inclusion criteria (see table 4) is unusually high compared with that which would be expected for the pathway as a whole. Many breast cancer pathways specifically ask for a panel of blood tests to be performed by GPs prior to 2WW referrals in males (for the investigation of gynaecomastia), which is not required for female referrals, suggesting bias.

We note that differences in the blood tests GPs are likely to provide in the lead up to/as part of a 2WW referral typically vary significantly depending on pathway. This is likely to be an important factor in explaining the difference in patient inclusion rates for each pathway we see for this work (see tables 1 and 2).

The choice to use blood measurements from up to 90 days prior to and up to 14 days postreferral is also a possible source of bias. Bloods taken significantly before referral can be biased because if the patient does have cancer, any tumour could be smaller or even not yet present at the time the blood test was administered. And bloods taken postreferral begin to run the risk that the decision was taken to order the blood test using

information not available at the time of referral. We have chosen this time frame as a reasonable balance between missing data and these potential biases. We note that for both values (90 days prior, 14 days post) we performed a sensitivity analysis during algorithm development where we varied these parameters and re-ran otherwise identical cross-validations. This showed that the choice of (90 days prior, 14 days post) was reasonably stable, and in particular, we did not see any significant gains in algorithm performance unless the post-referral cut-off was increased past 21 days, suggesting that while that source of bias does exist, it is not a significant factor with a 14-day postreferral cut-off.

### Implications for policy research and practice

Until we have undertaken a prospective evaluation of the performance of the algorithms it is not possible to predict how this will be used. However, we do envisage use of the tool, as part of clinical triage, to both prioritise those at higher levels of risk and de-prioritise those at the very lowest levels of risk, in conjunction with appropriate safety netting. We also need to fully understand the views of patients, clinicians, and commissioners on the acceptability and utility of the tests. We note that each 2WW pathway is distinct, with its own challenges and priorities, as well as differing prevalences of cancer (see, eg, Smith

**Table 6** Ninety per cent cancer rule-in

| Pathway | Proportion of non-cancers ruled-out (ie, not red-flagged) (specificity) (95% CI) | Positive predictive value (95% CI) |
|---|---|---|
| Breast | 0.4582 (0.4450 to 0.4715) | 0.0890 (0.0793 to 0.0991) |
| Lower Gastrointestinal | 0.2723 (0.2637 to 0.2811) | 0.0642 (0.0587 to 0.0697) |
| Upper Gastrointestinal | 0.3363 (0.3227 to 0.3503) | 0.0732 (0.0644 to 0.0822) |
| Gynaecological | 0.4674 (0.4473 to 0.4879) | 0.1134 (0.0972 to 0.1303) |
| Urological | 0.3548 (0.3379 to 0.3710) | 0.3044 (0.2878 to 0.3208) |
| Lung | 0.3625 (0.3238 to 0.3987) | 0.2541 (0.2178 to 0.2906) |
| Haematological | 0.4330 (0.3807 to 0.4849) | 0.4249 (0.3722 to 0.4759) |
| Head and neck | 0.2733 (0.2579 to 0.2885) | 0.0804 (0.0703 to 0.0911) |
| Skin | 0.3905 (0.3745 to 0.4068) | 0.3230 (0.3067 to 0.3392) |

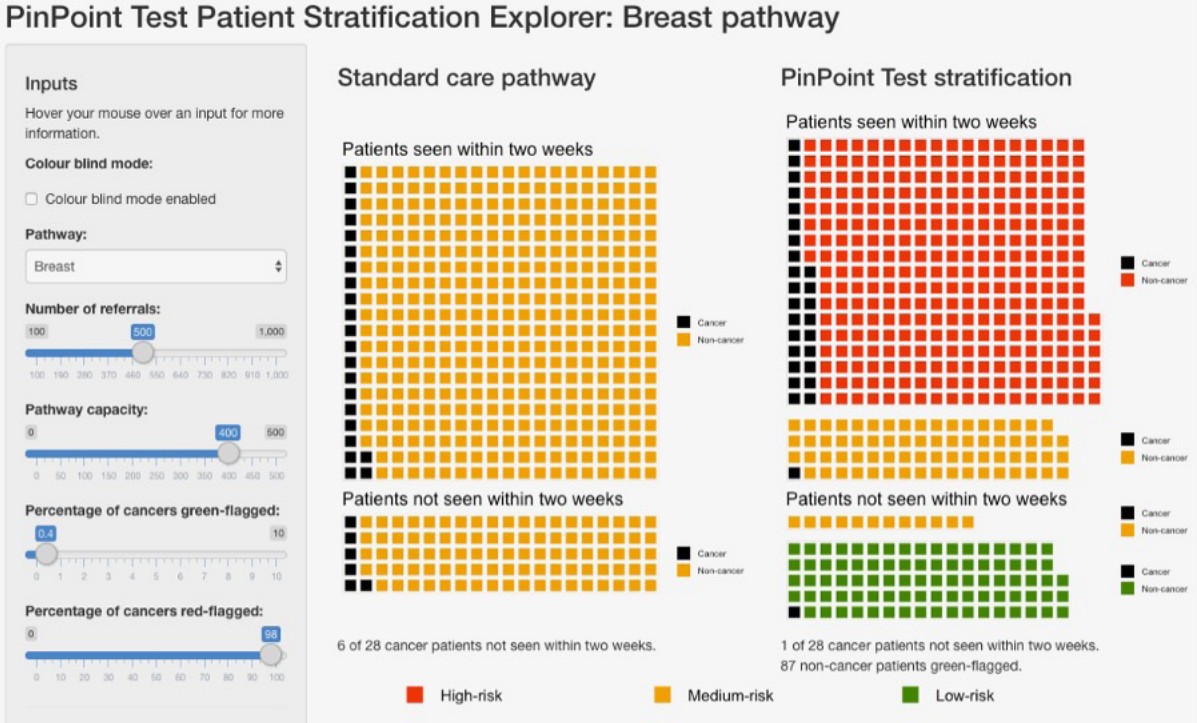

**Figure 2** Shows stratification of patients on the 2WW breast pathway using the relevant algorithm presented in this work, compared with the standard care pathway. given an urgent care pathway where the number of referrals exceeds the pathway capacity to see patients within 2 weeks, use of the test to stratify patients into risk categories (right) leads to a larger proportion of patients with cancer being seen when compared with the standard care pathway (left), in which patients are seen on a first-come, first-served basis. Patients highlighted in red are identified as being at high-risk for cancer (red-flagged), so can be expedited for further diagnostic testing. Patients highlighted in green are identified as being at very low risk for cancer (green-flagged), allowing for initial management in primary care rather than immediate referral to secondary care. The sliders on the left-hand side show the number of referrals, the number of patients that the pathway can handle in a given time frame (the pathway capacity), the percentage of cancers which are green-flagged (ie, setting a very low false negative rate, and therefore, high sensitivity c.f. table 5), and the percentage of cancers that are red-flagged (ie, identifying cases with high risk, so that they can be expedited for further diagnostic testing). The red-flagging slider effectively sets a sensitivity for the red-flagging process; setting sensitivity=0.9 corresponds to the results shown in table 6. The slider for 'percentage of cancers green-flagged' can be used to set the false negative rate and see the resulting performance of the test. collectively, this represents a possible approach to using the algorithms to improve the triage of patients referred to a 2WW pathway. An interactive version of this is available at https://www.pinpointdatascience.com/patient-test-stratification, we note that for the standard care pathway, all non-cancer patients are labelled in the same colour (yellow) to indicate that they are unstratified by the test. 2WW, 2-week wait.

*et al*[17]—these issues will likely require detailed consideration by all the key stakeholders on a pathway-by-pathway basis.

The 2WW pathways are an effective and well-used route for earlier cancer diagnosis in the NHS. However, the pressures resulting from this increased use and the current COVID-19 crisis mean that business-as-usual is no longer an option, and the NHS must adapt. New diagnostic technologies can be a part of this solution, giving clinicians better tools with which to triage patients and facilitate appropriate onward journeys through the healthcare system.

**Author affiliations**
[1]PinPoint Data Science Ltd, Leeds, UK
[2]University of Leeds, Leeds, UK
[3]NIHR MedTech and In Vitro Diagnostic Co-Operative, Leeds, UK
[4]University of Exeter, Exeter, UK
[5]Leeds Teaching Hospitals NHS Trust, Leeds, UK
[6]Chair of the PinPoint Scientific Advisory Board, Leeds, UK

**Acknowledgements** We thank the following people and organisations for their contributions: Emmylou Laird and Claire Eckert for their help with numerous aspects of this work. The Leeds Institute for Data Analytics, in particular Phil Chambers, for all their support in providing secure computing facilities. The pathology and health informatics staff of Leeds Teaching Hospitals Trust and Mid-Yorkshire Hospitals NHS Trust for their help. The X-Lab team for help with deployment into the NHS. The board members and partner organisations of the Leeds Centre for Personalised Medicine and Health and the Leeds Academic Health Partnership. The NIHR Leeds MIC and CRUK CANTEST PPI groups, especially Pete Whetstone and Margaret Johnson. We thank HealthWatch Leeds and Healthwatch Kirklees for patient consultation forums.

**Contributors** RS, MM, RDN, GH, RF and SD conceptualised the study, and led on the initial protocol development. GT, RF, NSa, BS and PS contributed towards funding applications and protocol refinement. RS, MDN, KLL and JRS developed the software and algorithms, performed the data analysis and completed the CE marking process, with clinical input from RN, SD, NSh, GH and PS and methodological input from BS, CJ and MM. GH led on the provision of deidentified data, assisted by CJ and RF. RF oversaw project management. All authors contributed to the interpretation of the results, writing of the manuscript and approved the final version. RS is the guarantor.

**Funding**  Aspects of this work have been supported by awards from MRC 'Proximity to Discovery' (MC_PC_17193), Local Enterprise Partnership (Leeds, 109550) and Innovate UK (33772). RDN, BS, GH and MM are funded by the NIHR Leeds In Vitro Diagnostic Co-operative (MIC-2016-015). PinPoint Data Science funded the data science work and time contributions of RS, MDN, KLL, JRS, GT, NS and RF. This research is linked to the CanTest Collaborative, which is funded by Cancer Research UK (C8640/A23385), of which RDN is an Associate Director, MM was a member of Senior Faculty, and BS was part funded.

**Competing interests**  RS, KLL, MDN, JRS, NS and GT are employed by and are shareholders in PinPoint Data Science. MM has been employed as a consultant to PinPoint Data Science in October to November 2020. Both the University of Leeds and Leeds Teaching Hospitals Trust have a royalty agreement with PinPoint Data Science, meaning that those institutions are likely to benefit financially in the event of PinPoint being commercially successful.

**Patient and public involvement**  Patients and/or the public were involved in the design, or conduct, or reporting, or dissemination plans of this research. Refer to the Methods section for further details.

**Patient consent for publication**  Not applicable.

**Ethics approval**  Data for the analysis are retrospective and fully deidentified before being released to the study team. The work was carried out under service evaluation with the formal approval of the Leeds Teaching Hospitals Trust R&I and Data Governance Committee (ref LTHT19020), and with the specific approval of the Trust Caldicott Guardian.

**Provenance and peer review**  Not commissioned; externally peer reviewed.

**Data availability statement**  The data will not be made available to others, as it is deidentified NHS patient data.

**ORCID iDs**
Richard Savage http://orcid.org/0000-0001-6025-1571
Richard D Neal http://orcid.org/0000-0002-3544-2744

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
