## [Reviewer comments · BMJ Open]

ARTICLE DETAILS

TITLE (PROVISIONAL)	Development and validation of multivariable machine learning algorithms to predict risk of cancer in symptomatic patients referred urgently from primary care
AUTHORS	Savage, Richard; Messenger, Mike; Neal, Richard; Ferguson, Rosie; Johnston, Colin; Lloyd, Katherine L; Neal, Matthew D; Sansom, Nigel; Selby, Peter; Sharma, Nisha; Shinkins, Bethany; Skinner, Jim R; Tully, Giles; Duffy, Sean; Hall, Geoff

VERSION 1 – REVIEW

REVIEWER	Malhotra, Ananya London School of Hygiene and Tropical Medicine Faculty of Epidemiology and Population Health
REVIEW RETURNED	13-Jul-2021

GENERAL COMMENTS	1. In Table 2, specify the component for which you have calculated prevalence. It seems you have calculated the prevalence only among cases with complete information on bloods, which does not reflect the true prevalence of cancer in the cohort. Please provide details of what you have presented.2. In Table 6, can you explain why some of the 2WW NHS pathways have such low PPV compared to others. Given that, can you suggest some measures to be taken into consideration by clinicians while interpreting these values.3. The number of non-cancer patients in the cohort are almost 5-6 times more than the number of cancer patients. How did you account for this imbalance in your data analysis?4. In Figure 1, under Analysis block of development set, how can you have 43650 cases in each of 8 pathways/referrals apart from breast cancers when total cases are 52028.5. The calculation of sample size in subsection 2.4 is not clear. How have you applied it in your analysis?6. Which software and version along with packages were used to carry out statistical analyses. Please include the programming codes in the supplementary files.7. In Figure 2, the colour coding is a bit misleading. If
--

	yellow is all patients at medium risk, then have you assumed that all non-cancer patients were at medium risk in standard care pathway before the ML algorithm was applied?  8. In Figure 2, the scale of 'Percentage of cancers green-flagged' should be from 0 to 100 and not 0 to 10. 9. In subsection 2.1, you mentioned that all patients from the 2WW pathway were included in the development set. Describe what is included in the 'all' other 2WW pathways apart from the nine you already considered. Are those cancers 'rare' in the population or just not enough cases were reported in LTHT. In the validation set you only included nine of the 2WW pathways. If ~98% of the referrals belonged to one of the nine pathways, then why not include only these nine pathways in the training set and merge all the remaining less frequent pathways as one label, i.e. the tenth 2WW pathway, renaming them as 'Other'. You can then have the same 10 labels in the validation set. It is not a convincing to exclude patients whose cancers are rare and loose out on the opportunity of prioritised diagnosis. How did you model in patients who had multiple referrals? 10. Can you elaborate subsection 2.7 and explain (while including relevant references) why you adopted this modelling strategy. Why you choose gradient boosting algorithms first and why and how the output of this model was used to fit a polynomial logistic regression model. Which gradient boosting algorithm did you use? Cite articles which explain the application of these models. Also explain briefly how you tuned the parameter and other metrics to avoid problems like overfitting. How the results from polynomial logistic regression model were applied in the PinPoint test application. 11. In Tables S1 and S2, explain 'Thresholds' for non-statistical audience. 12. For ease of readers, re-label some of the columns of Tables 5 and 6 as will be used in the PinPoint application. Or mention in the footnotes what values from the tables are plugged into the application. 13. Can you explain the utility and relevance of Figures S1, S2 and S3 with regards to your study.
--	---

REVIEWER	James, Jonathan Nottingham University Hospitals NHS Trust
-----------------	--

REVIEW RETURNED	01-Nov-2021
-------------

GENERAL COMMENTS	Interesting concept on using an AI algorithm to predict risk based on a panel of routinely available blood and then assessing the ability to triage patients to the two week wait cancer pathway. Abstract: The authors conclude that “the tests used are affordable and can be deployed rapidly to any NHS pathology laboratory with with no additional hardware requirements”. The authors give us no data on affordability and provide very little information on the blood tests being evaluated by the AI algorithm, and so this concluding sentence does not reflect the results presented and should be removed. Methods: Outcome: It is not clear exactly what the output of the algorithm is. Does the algorithm give a percentage chance of malignancy for each case, is it a binary output of ‘cancer’ or ‘no cancer’ or is it some other risk score. Could the authors clarify? Predictors: The description of the blood tests from each patient evaluated by the algorithm are minimal. I would expect more detail than just a “range of biochemistry measurements” or “standard tumour markers”. Are GPs asked to provide certain blood tests as part of the two week wait referral process? This may explain why the patient exclusions are considerably higher for some tumour pathways compared to others. I’m also not clear why you would include blood tests acquired up to 14 days post referral, presumably some of these will have been ordered by the cancer pathway having already seen the patient and may be influenced by the perceived malignancy risk following initial clinical assessment. I suspect the GPs may also order blood tests based on their initial clinical assessment of the risk of malignancy prior to referral. All these issues have considerable potential to introduce bias. It would be very interesting to see what the cancer incidence was in the patients excluded from the study due to lack of pre-referral blood tests compared to those included. Results: There is a rather large proportion of male patients meeting the inclusion criteria in the breast cancer pathway. 12% male patients would not be typical of a normal breast cancer two week wait referral profile. Again I suspect this may reflect bias over which patients are receiving pre-referral blood tests. I know many breast cancer pathways specifically ask for a panel of blood tests to be performed by GPs prior to two week wait referrals in males (for the investigation of gynaecomastia) which is not required for female referrals suggesting bias. Discussion: The discussion requires more detail on the potential sources of bias outlined above. It is only mentioned briefly as a bullet point in the limitations section and not discussed. Bias in patient selection and exclusions due to the performance of blood test either pre-referral or in the 14 days post referral has considerable potential to influence the results and affect the validity and usefulness of the findings and more detailed discussion is required. It may also mean that the algorithm is not actually generalisable to other patient populations or NHS Trusts where the criteria for performing pre-referral blood tests may be different.
---

VERSION 1 – AUTHOR RESPONSE

*Reviewer: 1
Dr. Ananya Malhotra, London School of Hygiene and Tropical Medicine Faculty of Epidemiology and Population Health*

In Table 2, specify the component for which you have calculated prevalence. It seems you have

calculated the prevalence only among cases with complete information on bloods, which does not reflect the true prevalence of cancer in the cohort. Please provide details of what you have Presented.

The prevalences listed in Table 2 are indeed for those cases which meet the acceptance criteria. We have added text to Table 2 to make this clear.

2. In Table 6, can you explain why some of the 2WW NHS pathways have such low PPV compared to others. Given that, can you suggest some measures to be taken into consideration by clinicians while interpreting these values.

The different 2WW pathways have very different prevalences (Table 1 shows this for the subsets of 2WW patients meeting the inclusion criteria for this work, and this is also the case for the 2WW pathways as a whole). Because of the relationship between PPV, NPV, sensitivity, specificity and prevalence, these very different prevalences tend to lead to very different PPVs and NPVs.

This is an important challenge for communicating results to clinicians. Our current thought on this is that it's very important to consider each 2WW pathway separately (this is also important because the pathways themselves are in some cases very different from each other). We have added some text to section 4.2 to note this, and a reference to a recent paper by Smith et al which looks at the varying referral, conversion, and detection rates across different 2WW pathways. We are grateful to the reviewer for raising this point, because it is an important consideration in how these tests can ultimately be used in clinical practice.

3. The number of non-cancer patients in the cohort are almost 5-6 times more than the number of cancer patients. How did you account for this imbalance in your data analysis?

The imbalanced classes are accounted for via upweighting the importance of the cancer patients in the gradient boosting algorithms. The same increased weight is applied to all cancer patients, and this is tuned as a hyperparameter during the development work (i.e. using cross-validation on the development set). We have added a paragraph to section 2.7 to this effect.

4. In Figure 1, under Analysis block of development set, how can you have 43650 cases in each of 8 pathways/referrals apart from breast cancers when total cases are 52028.

These numbers are correct, but we are grateful that the reviewer highlighted them as on reflection we had not described the origin of these numbers. They are the sizes of the training data set sizes used for the algorithm for each 2WW pathway. We discovered during development that modest performance gains could be achieved by using just the 2WW breast pathway data for the breast algorithm, and using the data for all other pathways for each of the other 8 algorithms (hence the same training data were used for all pathways except breast). We have added text to the legend for Figure 1 to explain this.

5. The calculation of sample size in subsection 2.4 is not clear. How have you applied it in your Analysis?

We thank the reviewer for highlighting this - we have revised the text in this subsection to clarify it. The sample size calculation was used to identify the 2WW pathways for which there was sufficient validation data to provide a meaningful validation. This is discussed under 'limitations' in the discussion section, but we have added a paragraph to subsection 2.4 to make this clearer.

6. Which software and version along with packages were used to carry out statistical analyses. Please include the programming codes in the supplementary files.

The code for this work is a set of 9 software IVD products (written in Python) which are CE marked and on the market in the UK. We regret that we are unable to provide this code, as it is commercially sensitive.

7. In Figure 2, the colour coding is a bit misleading. If yellow is all patients at medium risk, then have you assumed that all non-cancer patients were at medium risk in standard care pathway before the ML algorithm was applied?

All the non-cancer patients in the standard care pathway are labelled in the same colour (yellow) to indicate that they are unstratified by the test. We have added some text to the Figure 2 legend to make this clearer.

8. In Figure 2, the scale of 'Percentage of cancers green-flagged' should be from 0 to 100 and not 0 to 10.

We confirm that the scale is indeed 0 to 10 (i.e. correct as shown in Figure 2). This is because that particular slider is for the percentage of cancers which are green-flagged (i.e. one can use this slider to set a false negative rate and see the resulting performance of the test), and we felt that it was likely that a user would prefer finer control over very low values on this slider, rather than being able to set larger values >10. We have added some detail to the text accompanying Figure 2 to make this clearer, as we had not explained this point.

9. In subsection 2.1, you mentioned that all patients from the 2WW pathway were included in the development set. Describe what is included in the 'all' other 2WW pathways apart from the nine you already considered. Are those cancers 'rare' in the population or just not enough cases were reported in LTHT. In the validation set you only included nine of the 2WW pathways. If ~98% of the referrals belonged to one of the nine pathways, then why not include only these nine pathways in the training set and merge all the remaining less frequent pathways as one label i.e. the tenth 2WW pathway, renaming them as 'Other'. You can then have the same 10 labels in the validation set. It is not a convincing to exclude patients whose cancers are rare and loose out on the opportunity of prioritised diagnosis.

The 'all' other 2WW pathways are listed under 'limitations' in the discussion section, and also in the new text in subsection 2.4. They are: Testicular, Brain/CNS, Sarcomas, Children's Cancer, Acute Leukaemia, other cancer. These cancers are indeed rare in the population.

We included all pathways in the development set because our goal was to train algorithms that could assist with pan-cancer diagnosis, including cancer cases which have not been referred down the correct pathway. We chose not to combine all the less frequent pathways into a tenth 'other' 2WW pathway for validation because this would not correspond to any actual clinical pathway, and our goal in this work has been to develop algorithms that can be used directly in the NHS to help with the pressures being experienced by the 2WW pathways.

We have added some text to section 2.1 to note this.

How did you model in patients who had multiple referrals?

Patients who had multiple referrals are not modeled any differently than for any other patient. While information about repeated referral could, in principle, aid the algorithm, this would make the algorithm much harder to deploy in practice as it would need reliable access to an electronic healthcare record, rather than just being linked directly to the Laboratory Information Management System (LIMS) which handles the pathology lab data flows. We have therefore avoided this on practical grounds, for the time being.

We have added some text to section 2.1 to clarify this, and we're grateful to the reviewer for mentioning this as it's an important practical consideration to deploying such a technology in the NHS.

10. Can you elaborate subsection 2.7 and explain (while including relevant references) why you adopted this modelling strategy. Why you choose gradient boosting algorithms first and why and how the output of this model was used to fit a polynomial logistic regression model. Which gradient boosting algorithm did you use? Cite articles which explain the application of these models. Also explain briefly how you tuned the parameter and other metrics to avoid problems like overfitting. How the results from polynomial logistic regression model were applied in the PinPoint test application.

We have expanded subsection 2.7 to address these points. The parameters and other metrics are tuned using 5-fold cross-validation on the development set before being locked down. The purpose of the polynomial logistic regression model is to calibrate the predictions being generated by the gradient boosting algorithm - we have clarified this in the text, and added relevant references.

11. In Tables S1 and S2, explain 'Thresholds' for non-statistical audience.

We have added text to the supplementary materials to help explain 'thresholds' to a non-statistical audience.

12. For ease of readers, re-label some of the columns of Tables 5 and 6 as will be used in the PinPoint application. Or mention in the footnotes what values from the tables are plugged into the application.

We have revised the text accompanying Figure 2 to clarify the link between the PinPoint application and the Tables.

13. Can you explain the utility and relevance of Figures S1, S2 and S3 with regards to your study.

We have added text to these figures in the supplementary material to explain their utility and relevance.

Reviewer: 2

Dr. Jonathan James, Nottingham University Hospitals NHS Trust

Comments to the Author:

Interesting concept on using an AI algorithm to predict risk based on a panel of routinely available blood and then assessing the ability to triage patients to the two week wait cancer pathway.

We are pleased that the reviewer agrees that this is an interesting concept.

Abstract:

The authors conclude that “the tests used are affordable and can be deployed rapidly to any NHS pathology laboratory with with no additional hardware requirements”. The authors give us no data on affordability and provide very little information on the blood tests being evaluated by the AI algorithm, and so this concluding sentence does not reflect the results presented and should be removed.

This is an excellent point and we apologise for the oversight! We have added a table to the supplementary materials that gives a complete list of the analytes used by the algorithm, plus the 2018-19 NHS reference costs for those tests. We have also added a reference to a 2014 CRUK report ‘Saving Lives, Averting Costs’, which estimates per-patient referral costs to some 2WW pathways.

Methods:

Outcome: It is not clear exactly what the output of the algorithm is. Does the algorithm give a percentage chance of malignancy for each case, is it a binary output of ‘cancer’ or ‘no cancer’ or is it some other risk score. Could the authors clarify?

The algorithm outputs the probability (between zero and one) that the tested patient has a malignant neoplasm. The ‘Background’ section contains the following text on this point:

“We report the development and validation of a set of machine learning algorithms to provide a calibrated risk probability of cancer (a score between zero and one, higher values indicating greater risk of cancer) for triaging symptomatic patients”

We feel that this text is sufficient to explain the point, but we are happy to expand it if the reviewer thinks that further detail would be helpful.

Predictors: The description of the blood tests from each patient evaluated by the algorithm are minimal. I would expect more detail than just a “range of biochemistry measurements” or “standard tumour markers”.

We have added a table to the supplementary materials that gives a complete list of the analytes used by the algorithm, plus the 2018-19 NHS reference costs for those tests. This table also addressed the reviewer’s comment above about the abstract.

Are GPs asked to provide certain blood tests as part of the two week wait referral process? This may explain why the patient exclusions are considerably higher for some tumour pathways compared to others.

GPs are asked to undertake pre-referral blood tests for some 2WW pathways (e.g. Ca125 for suspected ovarian, and PSA for suspected prostate). This depends a lot on the pathway in question, and we agree that it is likely to be an important reason why there are differences between levels of patient exclusion in different pathways. We have added text in subsection 2.1 to note this and thank the reviewer for highlighting this, as it’s an important practical consideration for interpreting this work.

I’m also not clear why you would include blood tests acquired up to 14 days post referral, presumably some of these will have been ordered by the cancer pathway having already seen the patient and may be influenced by the perceived malignancy risk following initial clinical assessment. I suspect the GPs may also order blood tests based on their initial clinical assessment of the risk of malignancy prior to referral. All these issues have considerable potential to introduce bias.

This is a really important point, and one that we considered at length prior to undertaking the analysis. The choice to include bloods acquired up to 14 days post-referral was done to seek a reasonable balance between missing data and possible bias. We note the text from subsection 2.3 describing this (reproduced below). We have also added text in the Discussion section to explore this in more detail, and in particular we include that as part of the development work we performed a sensitivity analysis that showed that the inclusion of post-referral blood tests led to a significant increase in estimated test performance (i.e. likely bias) only if the cut-off was 21+ days post-referral. We had previously neglected to report this, so we’re grateful to the reviewer for highlighting such an important issue.

(from section 2.3)

“As a retrospective cohort, blood measurements were used where they were available in the database up to 90 days prior to referral or up to 14 days post referral. This was done to seek a reasonable balance between missing data and possible bias (for example if blood measurements were made after a diagnosis had been established). For example, it is risky to use blood measurements taken more than 14 days post-referral as there is an increasing chance that those

bloods could have been ordered by a clinician in response to a confirmed diagnosis of cancer. In routine clinical use, all model predictors would be available at the time.”

Current evaluation work in the NHS addresses this possible source of spectrum bias by testing all consenting patients in a given pathway (see also our response below to the final point)

It would be very interesting to see what the cancer incidence was in the patients excluded from the study due to lack of pre-referral blood tests compared to those included.

We have added Table S6 to the supplementary materials, containing overall prevalences for each pathway including patients who were excluded from the analyses. Interestingly, the use of cancer diagnoses up to 12 months after the referral date for a given patient does lead to somewhat inflated prevalence rates, relative to what one might expect for the 2WW pathways (see reference 17, Smith et al). For comparison, we've also included prevalences using only 2WW diagnoses made within 2 months of referral (corresponding to the NHSE 62 day target from referral to first treatment), which helps illustrate the cause of this inflation. We're grateful to the reviewer for highlighting this, as it's a useful additional insight into this source of possible spectrum bias.

Results: There is a rather large proportion of male patients meeting the inclusion criteria in the breast cancer pathway. 12% male patients would not be typical of a normal breast cancer two week wait referral profile. Again I suspect this may reflect bias over which patients are receiving pre-referral blood tests. I know many breast cancer pathways specifically ask for a panel of blood tests to be performed by GPs prior to two week wait referrals in males (for the investigation of gynaecomastia) which is not required for female referrals suggesting bias.

This is an important point. We were aware of the inclusion of male patients on the breast cancer pathway, but had not highlighted this adequately in the paper and in particular made no mention of the details described by the reviewer. We have added text under 'limitations' in the Discussion section of the paper to make this point, and we're grateful to the reviewer for highlighting this point and providing a very useful explanation.

Discussion: The discussion requires more detail on the potential sources of bias outlined above. It is only mentioned briefly as a bullet point in the limitations section and not discussed. Bias in patient selection and exclusions due to the performance of blood test either pre-referral or in the 14 days post referral has considerable potential to influence the results and affect the validity and usefulness of the findings and more detailed discussion is required. It may also mean that the algorithm is not actually generalisable to other patient populations or NHS Trusts where the criteria for performing pre-referral blood tests may be different.

We agree with these points. One of the significant challenges with this work is assessing the bias. We have significantly expanded the discussion of bias under 'limitations' in the Discussion section to explore this in greater detail.

The generalisability of the algorithm must, of course, be properly assessed. This work is currently underway as a service evaluation within the NHS, and we look forward to reporting those results in due course. This evaluation will also address the spectrum bias issue raised by the reviewer above. The service evaluation is listed under 'strengths' in the paper.

VERSION 2 – REVIEW

REVIEWER	James, Jonathan Nottingham University Hospitals NHS Trust
REVIEW RETURNED	14-Dec-2021

GENERAL COMMENTS	I note that the additions and clarifications requested following the previous review have been addressed. I have a few minor comments to improve the quality of the paper: Methods: 2.1: The last paragraph ("We note that differences in the blood tests") would be better included in the discussion section. Discussion: Summary of the main findings: This section would benefit from a re-write as it is duplicating too much of the results section and is not in the style I would expect from an opening section/paragraph of a discussion. The authors should provide a succinct summary of the entire study: restate briefly the background of the study, why it was done and then state major findings. The discussion would benefit from a concluding couple of sentences.
--

VERSION 2 – AUTHOR RESPONSE

We are very grateful to reviewer 2 for their additional comments. We have amended the manuscript in response to these comments and explain below the improvement we have made. In the marked copy of the paper we have used green highlighting to denote the new changes (the original changes being highlighted in blue).

>Reviewer: 2

>Dr. Jonathan James, Nottingham University Hospitals NHS Trust Comments to the Author:

>I note that the additions and clarifications requested following the previous review have been addressed. I have a few minor >comments to improve the quality of the paper:

>

>Methods: 2.1: The last paragraph ("We note that differences in the blood tests") would be better included in the discussion >section.

We agree and have moved this paragraph to the discussion section.

>Discussion: Summary of the main findings: This section would benefit from a re-write as it is duplicating too much of the results >section and is not in the style I would expect from an opening section/paragraph of a discussion. The authors should provide a >succinct summary of the entire study: restate briefly the background of the study, why it was done and then state major findings.

We have improved the discussion section in line with these comments. We have also removed the duplication of the results section, and moved one paragraph to the results section where it is better suited.

>The discussion would benefit from a concluding couple of sentences.

We have added some concluding sentences to better round out the discussion section.